# Oxidative Stress and Antioxidant Response in Populations of the Czech Republic Exposed to Various Levels of Environmental Pollutants

**DOI:** 10.3390/ijerph19063609

**Published:** 2022-03-18

**Authors:** Antonin Ambroz, Pavel Rossner, Andrea Rossnerova, Katerina Honkova, Alena Milcova, Anna Pastorkova, Jiri Klema, Jana Pulkrabova, Ondrej Parizek, Veronika Vondraskova, Jaroslav Zelenka, Nikola Vrzáčková, Jana Schmuczerova, Jan Topinka, Radim J. Sram

**Affiliations:** 1Department of Nanotoxicology and Molecular Epidemiology, Institute of Experimental Medicine CAS, Videnska 1083, 142 20 Prague, Czech Republic; anna.pastorkova@iem.cas.cz; 2Department of Genetic Toxicology and Epigenetics, Institute of Experimental Medicine CAS, Videnska 1083, 142 20 Prague, Czech Republic; andrea.rossnerova@iem.cas.cz (A.R.); katerina.honkova@iem.cas.cz (K.H.); alena.milcova@iem.cas.cz (A.M.); jan.topinka@iem.cas.cz (J.T.); radim.sram@iem.cas.cz (R.J.S.); 3Department of Computer Science, Faculty of Electrical Engineering, Czech Technical University in Prague, Karlovo Namesti 13, 121 35 Prague, Czech Republic; klema@fel.cvut.cz; 4Department of Food Analysis and Nutrition, Faculty of Food and Biochemical Technology, University of Chemistry and Technology, Prague, Technicka 3, 166 28 Prague, Czech Republic; jana.pulkrabova@vscht.cz (J.P.); ondrej.parizek@vscht.cz (O.P.); veronika.vondraskova@vscht.cz (V.V.); 5Department of Biochemistry and Microbiology, Faculty of Food and Biochemical Technology, University of Chemistry and Technology, Prague, Technicka 3, 166 28 Prague, Czech Republic; jaroslav.zelenka@vscht.cz (J.Z.); nikola.vrzackova@vscht.cz (N.V.); 6Department of Medical Genetics, L. Pasteur University Hospital, Trieda SNP 1, 040 11 Kosice, Slovakia; genetikaodd.snp@unlp.sk

**Keywords:** oxidative damage, DNA, lipids, antioxidant response, environmental factors, POPs

## Abstract

We aimed to identify the variables that modify levels of oxidatively damaged DNA and lipid peroxidation in subjects living in diverse localities of the Czech Republic (a rural area, a metropolitan locality, and an industrial region). The sampling of a total of 126 policemen was conducted twice in two sampling seasons. Personal characteristics, concentrations of particulate matter of aerodynamic diameter <2.5 µm and benzo[a]pyrene in the ambient air, activities of antioxidant mechanisms (superoxide dismutase, catalase, glutathione peroxidase, and antioxidant capacity), levels of pro-inflammatory cytokines (TNF-α, IL-1β, and IL-6), concentrations of persistent organic pollutants in blood plasma, and urinary levels of polycyclic aromatic hydrocarbon metabolites were investigated as parameters potentially affecting the markers of DNA oxidation (8-oxo-7,8-dihydro-2′-deoxyguanosine) and lipid peroxidation (15-F2t-isoprostane). The levels of oxidative stress markers mostly differed between the localities in the individual sampling seasons. Multivariate linear regression analysis revealed IL-6, a pro-inflammatory cytokine, as a factor with the most pronounced effects on oxidative stress parameters. The role of other variables, including environmental pollutants, was minor. In conclusion, our study showed that oxidative damage to macromolecules was affected by processes related to inflammation; however, we did not identify a specific environmental factor responsible for the pro-inflammatory response in the organism.

## 1. Introduction

Numerous studies have confirmed the negative role of environmental pollution in the development of many diseases, including those affecting, for example, the cardiovascular [1], pulmonary [2], excretory [3], or nervous [4] systems. The association of ambient air exposure with cancer risk resulted in the classification of air pollution as human carcinogen by the International Association for Research on Cancer (Group 1) [5]. Air pollutants include a complex mixture consisting of gases (volatile organic compounds, NOx, CO, and ozone), and particulate matter (PM) of various aerodynamic diameter and the chemicals bound to it (notably metals and organic compounds, such as polycyclic aromatic hydrocarbons, PAHs; or persistent organic compounds, POPs).

The biological effects of PM on the human organism depend on its physico-chemical properties: concentration, size, shape, and chemical composition, as well as the presence of organic and inorganic substances bound to its surface. The inhalation of fine PM (PM of aerodynamic diameter <2.5 µm; PM_2.5_) has been repeatedly shown to contribute to the development of asthma, chronic obstructive pulmonary disease, and cardiovascular diseases [6]. PM_2.5_ can penetrate into the lung parenchyma and interstitium, where it is deposited [7]. Additionally, it may enter the bloodstream and spread to other potentially sensitive organs, such as the bone marrow and lymph nodes [8]. The first protection against infiltrating PM_2.5_ in the lower respiratory tract is mediated by a non-specific immune response involving phagocytic alveolar macrophages, during which pro-inflammatory and other mediators are released and reactive oxygen species (ROS) are formed. ROS are then associated with oxidative stress induction in the lungs [9]. The underlying mechanisms by which pulmonary oxidative stress leads to systemic inflammation in response to air pollution is still not fully understood. It is assumed that proinflammatory mediators released during PM phagocytosis in the pulmonary alveoli may enter the bloodstream and cause systemic inflammation and oxidative stress [10]. Apart from phagocytic processes, PM_2.5_ contributes to ROS production due to the presence of compounds with pro-oxidant properties, e.g., PAHs and POPs. PAHs are mostly formed by the incomplete combustion of organic material. Some of these are possibly carcinogenic to humans; benzo[a]pyrene (B[a]P) has been identified as a human carcinogen [11]. One of the pathways of PAH metabolic activation involves the formation of o-quinones and ROS generation, thus contributing to oxidative stress [12]. POPs represent a broad class of chemicals that have been used in industrial applications, pest and disease control, or to increase crop production. Similar to PAHs, they bind to the aryl hydrocarbon receptor, and their metabolism involves CYP1A1 activation subsequently leading to ROS production [13].

Oxidative stress is a complex process that arises as a consequence of an imbalance between the levels of ROS and antioxidant defenses in an organism [14]. Cells are protected against oxidative damage by the activities of antioxidant enzymes (e.g., superoxide dismutase (SOD), catalase (CAT) and glutathione peroxidase (GPx)), along with the small antioxidant molecules mostly supplemented from dietary intake (vitamin C, vitamin E, carotenoids, and other compounds) [15]. ROS are necessary for the regulation of signaling pathways involved in cell growth, proliferation, differentiation, and survival [16]. However, if the capacity of antioxidant mechanisms is insufficient, ROS induce kinases leading to the activation of transcription factors (NF-κB, AP-1), propagating the pro-inflammatory signaling cascade and release of cytokines (e.g., TNFα, IL-1β, and IL-6), chemokines and other inflammatory molecules [17].Additionally, ROS interact with cellular macromolecules and cause their damage. Oxidative damage to DNA is mostly induced by the attack of ROS on nucleobases. If not repaired, the oxidized nucleobases may induce mutations. 8-oxo-7,8-dihydro-2′-deoxyguanosine (8-oxodG), a predominantly formed oxidized nucleotide that represents the rate of oxidation of guanine in the nucleotide pool, is excreted to urine [18]. Its levels in spot urine samples may be used as a biomarker of short-term exposure to air pollution [19]. An ROS-mediated attack on cell membrane lipids, such as polyunsaturated fatty acids, (PUFA) modifies cell membrane properties resulting in the disruption of regular cellular functions [20]. Peroxidation of arachidonic acid (AA), a polyunsaturated fatty acid abundantly contained in cell membranes independent of cyclooxygenases, leads to the formation of a number of products that include isoprostanes (IsoPs) [21]. IsoPs are cleaved from the sites of origin and then either circulated in plasma or excreted in urine [22]. Quantification of 15-F2t-isoprostane (15-F2t-IsoP) is considered a reliable index of the oxidative stress status in vivo. Although urine collection is an easily available and non-invasive method, the determination of plasma 15-F2t-IsoP provides a more accurate view of the overall oxidative damage of an organism [23].

As oxidative stress is generally accepted as a key condition involved in the pathogenesis of many diseases [24], it is, therefore, of utmost importance to investigate the factors that contribute to the induction of oxidative damage of macromolecules. To address these issues, we conducted a comprehensive study in which we aimed to identify the impact of selected parameters of individual characteristics, antioxidant and immune response, and the environmental pollution on markers of oxidatively damaged DNA (8-oxodG) and peroxidized lipids (15-F2t-IsoP). The study, that involved policemen working in three cities of the Czech Republic, was performed across two sampling seasons that were expected to differ in environmental pollution levels. To account for pro-oxidant and antioxidant processes in the study subjects, and compounds affecting oxidative stress, we further analyzed the antioxidant capacity, activities of antioxidant enzymes, and levels of selected inflammatory markers, as well as the plasma levels of selected POPs and urinary concentrations of PAH metabolites.

## 2. Materials and Methods

### 2.1. Subjects and Sampling

The cohorts originated from three geographically distant localities of the Czech Republic with specific local characteristics: Ceske Budejovice (CB, a regional center in a rural area, 16 subjects), Prague (the capital city of the Czech Republic, 56 subjects), and Ostrava (the center of industrial production characterized by high air pollution levels, 54 subjects) [25] (Appendix A). The samples (venous blood and urine) were collected from healthy non-smoking policemen at the end of their shifts. The sample collection was conducted in two sampling periods: in winter/early spring 2019 (Season 1, 4–7 March, 24–29 March, and 10–15 March, for CB, Prague, and Ostrava, respectively), when elevated air pollution levels were expected, and in early autumn 2019 (Season 2, 8–10 October, 20–25 October, and 29 September–4 October, for CB, Prague, and Ostrava, respectively), the season usually characterized by lower concentrations of pollutants. Each subject completed a personal questionnaire on medical history, socio-economic factors and lifestyle. Smokers and subjects with medical treatment were not included in the study. The participants were identical in both sampling periods. Blood was drawn to EDTA tubes to isolate plasma. Urine samples were collected into 50 mL tubes (Greiner Bio-one, Kremsmünster, Austria) and kept in aliquots (2 mL) at −80 °C until analysis of creatinine, cotinine and 8-oxodG. Samples of plasma, used for analysis of antioxidant enzymes (SOD, CAT, GPx), pro-inflammatory cytokines (IL-1β; IL-6), and 15-F2t-IsoP, were stored at −80 °C. All participants signed an informed consent form and could cancel their participation at any time, according to the Helsinki II Declaration. The study was approved by the Ethics Committee of the Institute of Experimental Medicine CAS in Prague.

### 2.2. Air Pollution Exposure Monitoring

Individual exposure to B[a]P was measured by personal monitors used by the study subjects during a 24 h period before blood sample collection. The active PV 1.7 monitors (URG, Chapel Hill, NC, USA) were equipped with Teflon-impregnated glass fiber filters T60A20 (Pallflex) collecting PM_2.5_ particles. The sampler was connected to a pump operating at 1.7 L/min powered by batteries with an inlet attached to the individual’s breathing zone and was located by his bed during the night. The detailed information regarding the air sampling was previously described [26]. Quantitative chemical analysis of B[a]P was performed by gas chromatography, coupled with tandem mass spectrometry, in electron ionization. Concentrations of B[a]P were expressed in ng/m^3^.

Information on the ambient air concentrations of PM_2.5_ and ozone during the sampling periods was obtained from the database of the Czech Hydrometeorological Institute (www.chmi.cz, (accessed on 9 March 2022)), the national authority responsible for monitoring the levels of environmental pollutants. The study subjects lived and worked within the range of the sampling sites.

### 2.3. Quantification of Persistent Organic Pollutants

Persistent organic pollutants included polychlorinated biphenyls (PCB), organochlorinated pesticides (dichlorodiphenyldichloroethylene, DDE; dichlorodiphenyldichloroethane, DDD; dichlorodiphenyltrichloroethane, DDT; hexachlorobenzene, HCB; and hexachlorocyclohexanes, HCH), brominated flame retardants (decabromodiphenyl ethers, BDE), and perfluoroalkylated substances (perfluoro-1-butanesulfonate, PFBS; perfluoro-1-hexanesulfonate, PFHxS; perfluoro-1-octanesulfonate, PFOS; perfluoro-1-decanesulfonate, PFDS; perfluoro-n-butanoic acid, PFBA; perfluoro-n-heptanoic acid, PFHpA; perfluoro-n-octanoic acid, PFOA; perfluoro-n-nonanoic acid, PFNA; perfluoro-n-decanoic acid, PFDA; perfluoro-n-undecanoic acid, PFUdA; perfluoro-n-dodecanoic acid, PFDoA; perfluoro-n-tridecanoic acid, PFTrDA; and perfluoro-n-tetradecanoic acid, PFTeDA). The compounds were assessed in plasma samples using gas chromatography coupled to (tandem) mass spectrometry [GC–MS/(MS)], and ultra-high performance liquid chromatography coupled to triple quadrupole tandem mass spectrometry (UHPLC–MS/MS), as previously described in detail [27]. The results were expressed in ng/mL plasma (for polar POPs) or ng/g of lipid weight (for non-polar compounds).

### 2.4. Analyses of PAH Metabolites in Urine

Monohydroxylated PAH metabolites (naphthalene-1-ol, 1-OH-NAP; naphthalene-2-ol, 2-OH-NAP; fluorene-2-ol, 2-OH-FLUO; phenanthrene-1-ol, 1-OH-PHEN; phenanthrene-2-ol,2-OH-PHEN; phenanthrene-3-ol, 3-OH-PHEN; phenanthrene-4-ol, 4-OH-PHEN; phenanthrene-9-ol, 9-OH-PHEN; and pyrene-1-ol, 1-OH-pyrene) were assessed in urine using tandem mass spectrometry. The concentration of the compounds was normalized per creatinine content (ng/g creatinine).

### 2.5. Cotinine Assay

Urinary cotinine concentrations as a marker of active and passive smoking were determined by the radioimmunological method as described [28]. The method is based on the competition between radiolabeled and unlabeled cotinine, for binding with a limited number of specific antibody binding sites. The analysis of each sample (diluted with water according to the sample concentration) was performed in technical duplicate using a 10 μL sample/well. Urinary cotinine levels were expressed in ng/mg creatinine.

### 2.6. Creatinine Assessment

A Jaffe method based on the reaction with picric acid [29] was used to determine the levels of creatinine in urine samples. The samples were measured in technical triplicate at 490 nm, 50 µL sample/well, and creatinine concentrations were expressed in mmol/L.

### 2.7. Analyses of Antioxidant Enzyme Activities

#### 2.7.1. Superoxide Dismutase Activity

The method is based on the competition of superoxide between disproportionation to oxygen and water by the activity of superoxide dismutase (SOD), or the reaction with nitroblue tetrazolium (NBT) yielding a color product. Superoxide is generated by the reaction between xanthine and xanthine oxidase. Briefly, a mixture of 100 μM xanthine, 250 μM NBT, 100 μM EDTA, and plasma samples diluted 100× in phosphate buffer was prepared. From this solution, 80 μL was pipetted in triplicate into a 37 °C tempered 96-well plate, and a reaction was started by adding 20 μL of xanthine oxidase diluted in phosphate buffer (resulting enzyme activity: 0.2 mU/mL). The kinetics of the reaction was measured for 1.5 h at 560 nm and 37 °C. A calibration curve of the enzyme activities was used to determine the activity of SOD. The results were expressed in U/mL.

#### 2.7.2. Catalase Activity

In the reaction, colorless ammonium heptamolybdate tetrahydrate is converted to a yellow product by the reaction with hydrogen peroxide. The resulting absorbance is inverse to the activity of catalase (CAT). Briefly, ammonium heptamolybdate tetrahydrate was dissolved in 0.025 M sulfuric acid. The CAT reaction was initiated by mixing hydrogen peroxide (50 mM solution in phosphate buffer) with 100× diluted plasma; 80 µL of the mixture was pipetted in triplicate into a 96-well plate and incubated for 1 h at 37 °C. The remaining hydrogen peroxide was then visualized by the addition of 20 μL of a 4% molybdate solution; the absorbance was measured at 360 nm and a reference wavelength of 600 nm. A calibration curve was used to determine the catalase activity in the plasma samples. The results were expressed in U/mL.

#### 2.7.3. Glutathione Peroxidase Activity

In the reaction, reduced glutathione is converted to its oxidized form by the activity of glutathione peroxidase (GPx). Glutathione is subsequently regenerated by added glutathione reductase at the expense of NADPH. The decrease of NADPH absorbance is therefore directly proportional to GPx activity. The reaction was performed in 100× diluted plasma samples using a commercial kit (Trevigen, Gaithersburg, MD, USA), according to the manufacturer’s recommendations. The results were expressed in U/mL.

#### 2.7.4. Oxygen Radical Absorbance Capacity (ORAC)

This method was used to analyze the antioxidant activity of the samples. In the reaction, a fluorescent probe (dipyridamole) is quenched by reactive oxygen species, generated from 2,2′-azobis(2-methylpropionamidin)dihydrochloride (AAPH). The resulting decrease in fluorescence is further modulated by the presence of antioxidants (antioxidant capacity) in the sample. Briefly, plasma samples were diluted in 2.5 µM dipyridamole and phosphate buffer to a final dilution of 100×. The solution (80 µL) was added in triplicate to a 96-well plate and the reaction was started by pipetting 20 µL of 100 mM solution of AAPH. The fluorescence was measured during a period of 2 h at an excitation wavelength of 415 nm and emission wavelength of 480 nm at 37 °C. A calibration curve of Trolox, an analog of vitamin E, was used to determine ORAC, expressed in µM Trolox Equivalent.

### 2.8. Analysis of Cytokines

The production of selected cytokines (TNFα, IL-1β, IL-6) in plasma was assessed by commercial ELISA kits (kit for TNFα from R&D Systems, Minneapolis, MN, USA; kit for IL-1β from Boosterbio, Pleasanton, CA, USA; kit for IL-6 from BioLegend, San Diego, CA, USA) according to the manufacturer’s instructions. Concentrations of cytokines were expressed in pg/mL.

### 2.9. Analysis of Oxidative Damage Markers

#### 2.9.1. 8-oxodG ELISA

First, urine samples were purified by solid phase extraction (SPE) as described in Rossner et al. [30] and then urinary 8-oxodG levels were determined by a competitive ELISA. The Highly Sensitive 8-OhdG Check ELISA (JaICA, Shizuoka, Japan) was performed according to the manufacturer′s recommendations with some modifications. Purified urine samples (diluted 1:2 with PBS) were incubated with primary antibody (N45.1) at 4 °C overnight and the following day each sample was analyzed in technical duplicate using a 50 µL sample/well. Absorbance was measured with SpectraMax®iD3 (Molecular Devices, San Jose, CA, USA) at 450 nm, and urinary 8-oxodG concentration was expressed as nmol 8-oxodG/mmol of creatinine.

#### 2.9.2. 15-F2t-IsoP ELISA

Blood plasma sample purification and plasma 15-F2t-IsoP analysis were performed according to the protocol for 8-isoprostane ELISA kit from Cayman Chemical Company (Ann Arbor, MI, USA). Each sample (125 μL) was first hydrolyzed and further purified using 8-Isoprostane Affinity Sorbent (Cayman, Ann Arbor, MI, USA). Following evaporation of the elution solution, purified samples were dissolved in ELISA buffer (330 μL) and analyzed in technical duplicates using 50 μL sample/well. Absorbance was measured by SpectraMax^®^iD3 (Molecular Devices, San Jose, CA, USA) at 405 nm, and plasma 15-F2t-IsoP concentration were expressed as pg 15-F2t-IsoP/mL plasma.

### 2.10. Statistical Analysis

Statistical calculations were performed using GraphPad Prism 9.3.1 (San Diego, CA, USA) and IBM SPSS Statistics for Windows, Version 20.0software (IBM Corp., Armonk, NY, USA). The normality of distribution was tested using the Kolmogorov–Smirnov test. The data distributed normally were further processed using *t*-test and ANOVA, while for the non-normally distributed data, Mann–Whitney and Kruskal–Wallis tests were applied. The analyzed parameter levels were assessed for differences between individual sampling periods in either locality, and for differences between localities within a sampling period. Multicollinearity between the variables of environmental pollution (POPs and PAHs metabolites) and oxidative stress markers was controlled using the variance inflation factor. Parameters showing a high degree of correlation were removed from further analysis; they included PCB 138, PCB 153, PCB 170; BDE 183; PFNA, PFDoA, PFTrDA, and PFTeDA. Associations of the levels of environmental pollutants, parameters of antioxidant response, and inflammation with oxidative stress markers (8-oxodG, 15-F2t-IsoP) were studied using generalized estimating equations (GEE). This approach is applied for correlated data that often appear in longitudinal studies, where tested parameters are measured in the same subjects at different time points. In the first step, bivariate comparisons between oxidative stress markers and individual studied variables were performed; the resulting *p*-values were corrected for multiple comparisons using the Benjamini–Hochberg method [31]. Multivariate analyses were adjusted to personal characteristics, antioxidant and immune response parameters, and those environmental pollutants for which significant associations were found in bivariate comparisons. The results of the multivariate analyses were visualized as scatter plots in which dependent variables (levels of 8-oxodG and 15-F2t-IsoP) were plotted against predicted B values calculated from B coefficients of variables included in GEE models.

## 3. Results

The characteristics of the study populations and a comparison of parameters of environmental pollution, antioxidant and immune response, and oxidative stress in both sampling seasons for individual localities are presented in Table 1, Table 2 and Table 3 and Appendix A. In CB, a rural locality, no difference in the ambient air levels of B[a]P and PM_2.5_ was detected; concentrations of ozone were significantly elevated in Season 1 (Table 1). No differences between seasons were noted for most of the analyzed urinary PAH metabolites. In contrast, the plasma levels of some POPs, particularly polychlorinated biphenyls, e.g., PCB 138, 153, 170, and 180, were elevated in samples collected in Season 2 (Appendix A). Additionally, antioxidant and immune response parameters tended to be higher in Season 1. The analysis of oxidative stress markers did not reveal a consistent result: in samples collected in Season 1, oxidative damage to DNA was elevated, while lipid peroxidation in this season was significantly lower than in Season 2 (Table 1). Subjects from Prague, a metropolitan locality, tended to be exposed to elevated concentrations of B[a]P in Season 2 (Table 2), although this result was not reflected in urinary PAH metabolite concentrations; ozone levels were higher in Season 1. Similar to subjects from the rural locality, the plasma levels of POPs were rather higher in samples obtained in Season 2 (Appendix A). Antioxidant and immune response in these subjects were not consistent across the seasons, while lipid peroxidation was elevated in the samples collected in Season 1 (Table 2). In Ostrava, the industrial locality, non-consistent significant differences in levels of environmental air pollutants were detected in individual seasons (Table 3). A similar result was observed for urinary PAH metabolite levels, although for most of the analyzed markers the differences were not significant. The plasma POPs concentrations followed a trend comparable with other localities (Appendix A). Importantly, it should be noted that the number of individual POPs, for which significant differences between the localities were detected, increased in the order CB-Prague-Ostrava. The antioxidant response and IL-1β production in the Ostrava samples was higher in those collected in Season 1; no differences in the levels of oxidative stress between the sampling seasons were detected (Table 3).

The results of the comparison between the localities in individual sampling seasons are reported in Table 4. The subjects did not differ by age or BMI. For education and for cotinine, a marker of tobacco smoke exposure, a difference was detected in Season 2. While the study participants reported to be non-smokers, these results most likely reflect passive smoking for some individuals in the Ostrava region. All parameters of antioxidant activities differed between the regions in Season 1; in Season 2 no difference was detected for SOD and ORAC. The levels of inflammatory markers were mostly comparable, with some non-consistent differences for IL-1β and IL-6. The plasma levels of some POPs differed between the localities, regardless of the sampling season, e.g., PCB 153, 170, 180, p,p′-DDD, BDE 47, PFNA, or PFDA. For others, the difference was detected only in one sampling season (e.g., PCB 52, 101, 138, o,p′-DDT, p,p′-DDT, or BDE 209). However, for almost 45% of the analyzed POPs (17 of 38), no difference between the localities was found for either sampling season. It is interesting to note that the urinary levels of most of the PAH metabolites differed between localities in both (5 of 9), or at least one (2 of 9) sampling season, suggesting differences in the total PAH exposure between the studied populations.

An overview of the results of bivariate analyses of association between oxidative stress markers and parameters of environmental pollution, antioxidant response, and inflammation is presented in Appendix A. Thereafter, the correction for multiple comparisons of plasma levels of IL-6 were identified to be positively associated with the marker of both oxidatively damaged DNA and peroxidized lipids. For 8-oxodG, we found negative associations with the plasma levels of o,p′-DDE and BDE 154. The levels of 15-F2t-IsoP were negatively correlated with concentrations of BDE 154, while the association with BDE 99 was positive. Lipid peroxidation was further affected by the locality–subjects from the Ostrava region tended to have increased plasma concentrations of 15-F2t-IsoP, than the study participants from other localities.

Multivariate estimates of associations between potentially confounding factors, including POPs, that were significantly correlated with oxidative stress markers in bivariate analyses, and parameters of oxidatively damaged DNA and lipid peroxidation, are reported in Table 5, and Figure 1 and Figure 2. Urinary 8-oxodG levels increased with decreasing antioxidant capacity (ORAC) and increasing plasma concentration of IL-6, a pro-inflammatory cytokine. In addition, a borderline positive association of GPx activity with 8-oxodG was noted. The effect of environmental pollutants on oxidatively damaged DNA was minor, as concentrations of only two compounds (o,p′-DDE, BDE 154) were found to be associated with the excretion of 8-oxodG to urine. Lipid peroxidation was influenced by the activity of CAT and plasma concentration of IL-6. Exposure to tobacco smoke was another parameter associated with the levels of 15-F2t-IsoP. Finally, the effects of POPs were identified: BDE 154 was negatively linked with lipid peroxidation, while for plasma BDE 99 levels, a positive correlation was observed.

## 4. Discussion

In the organism, cellular pro-oxidant and antioxidant mechanisms are responsible for maintaining the redox balance. The balance might be affected by various endogenous and exogenous factors, including, for example, biochemical processes in the organism, genetic background, lifestyle (physical activity, smoking, alcohol drinking, diet), age, occupation, or exposure to environmental pollutants. These factors create a complex network of interactions that influence the parameters characterizing oxidative stress levels in the organism. In populations residing in diverse localities of the Czech Republic, we aimed to identify variables with the most significant impacts on markers of oxidatively damaged DNA and lipid peroxidation. The localities differed both geographically and by prevailing industrial/agricultural activities. This experimental design allowed us to investigate a wide range of parameters potentially affecting oxidative stress in the study subjects, policemen spending most of their shifts outdoors. Among other factors, the impact of POPs was investigated. To the best of our knowledge, the association of plasma concentrations of POPs with oxidative stress markers has never been studied on such a large scale. Overall, our approach helped us to identify the possible role of environmental factors in oxidative stress induction.

The measurement of environmental pollutants in individual seasons across the localities showed significant differences in air pollution levels (Table 4). In Season 1, Ostrava was the most polluted locality when personal exposure to B[a]P was considered. We assume that, as a result of exposure, chronic inflammation may have developed in the subjects residing in the city, as suggested by a recent study [32]. This is supported by the highest levels of IL-1β detected in the plasma of Ostrava policemen in Season 1 (Table 3). Interestingly, while in Season 2 B[a]P exposure was highest in Prague, pro-inflammatory response represented by IL-6 plasma concentrations was most pronounced in the subjects from Ostrava (Table 3). This observation suggests that residents in this locality may be affected by chronic inflammation caused by long-term exposure to high levels of environmental pollutants. The results of antioxidant enzyme activities indicate fluctuations between seasons for the subjects from all localities. However, the antioxidant defense is mediated not only by the enzymes, but also by other mechanisms that were not investigated in this study [33], and that might explain some of our observations. These mechanisms include nonenzymatic systems (such as vitamin E, vitamin C, carotenes, ferritin, reduced glutathione, flavonoids, coenzyme Q, bilirubin, or cysteine that act as electron donors, thus neutralizing ROS) and repair enzymes responsible for elimination of molecules damaged by the attack of ROS (e.g., DNA repair enzymes or proteolytic enzymes). In Season 1, the highest 15-F2t-IsoP levels were found in the Ostrava subjects when compared with other localities, although a significant difference was detected only for the samples from CB. This result suggests that oxidative stress increases as a consequence of the highest B[a]P exposure in the Ostrava samples. In Season 2, the highest 8-oxodG levels were detected in the Prague subjects, which corresponds to the elevated concentrations of pollutants in Prague in this season. It should be further noted that exposure to ozone, highest in Prague, particularly in Season 1, was not associated with elevated oxidative stress response in subjects from any locality.

Similar to B[a]P and ozone, the concentrations of POPs in blood plasma also differentiated in the subjects from individual sampling localities and sampling seasons. The most pronounced variability was detected for polychlorinated biphenyls (about 50% of the analyzed compounds), while the concentrations of organochlorinated pesticides differed between the localities rather in Season 1. For brominated flame retardants and per/polyfluoroalkylated substances, the seasonal profiles were similar, and differences were observed for about 30% of the compounds.

The observations reported above suggest differences between the sampling seasons and/or the study populations. Using multivariate regression analysis, we aimed to identify the variables that significantly contributed to oxidative stress induction. Personal characteristics investigated in our study were not associated with either marker of oxidative stress. Cotinine levels were the only exception; this marker of cigarette smoke exposure negatively affected lipid peroxidation. However, this observation was driven by outlying cotinine values, most likely linked with passive smoke exposure and/or misinformation on smoking status reported by the study subject and should be regarded with caution.

Antioxidant mechanisms in the study subjects were assessed by the analysis of SOD, CAT, and GPx activities and antioxidant capacity of the blood plasma, evaluated using the ORAC assay. The induction of these mechanisms should contribute to the decreased levels of oxidative stress markers [34]. While we were able to identify ORAC as a parameter that is negatively associated with urinary excretion of 8-oxodG, increased activities of GPx and CAT were linked with elevated levels of 8-oxodG and 15-F2t-IsoP, respectively. Both enzymes catalyze the conversion of hydrogen peroxide to water, GPx reduces lipid peroxides to corresponding alcohols, protecting the cells against oxidative stress. Thus, our observation is unexpected, particularly considering the results of others who reported decreased activities of antioxidant enzymes to be accompanied by the induction of oxidative damage [35,36,37,38]. The resulting oxidative damage is affected, among other factors, by the sufficient capacity of antioxidant enzymes to cope with pro-oxidant insults in the organism. We speculate that in our study subjects the enzymes were induced, but their activity was not able to compensate oxidative stress already present in the organism. Increased levels of both the activities of antioxidant enzymes and oxidative stress markers may, thus, indicate chronic inflammation in the organism.

Markers of oxidative stress could be further affected by inflammatory processes in the organism. We analyzed the plasma levels of TNF-α, IL-1β, and IL-6 and found that the levels of oxidatively damaged DNA, as well as lipid peroxidation, were positively associated with IL-6. This cytokine, produced by macrophages, acts as a pro-inflammatory molecule whose effect may contribute to the induction of oxidative stress [39].

We further studied the potential effect of environmental contaminants on markers of oxidative damage. Unexpectedly, no significant associations of selected air pollutants (PM_2.5_, B[a]P) with either marker of oxidative stress were found. Both factors have previously been shown to be associated with oxidative damage [40,41], although negative results have also been reported [42,43]. While POPs have been shown to contribute to oxidative stress [36,44], for most of the studied molecules, we did not detect any significant effects. We observed a positive association between the plasma concentrations of BDE 99 and 15-F2t-IsoP; however, for other compounds (BDE 154, o,p′-DDE), significant negative correlations were detected. Polybrominated diphenyl ethers (PBDEs) are classified as semi-volatile organic compounds (SVOCs) that present health risks because they tend to accumulate at fine, inhalable particle fraction (PM_2.5_), rather than at a coarse one [45]. Therefore, we may assume that exposure to PBDE bound to fine particulate matter, may be related to the induction of oxidative stress starting in the alveoli. The association of PBDE inhalation with health risks was also confirmed by another study [46], that focused on the exposure to PBDEs in an indoor environment. In most situations; however, it is difficult to specify the exact mechanism of the action of POPs. The processes that these compounds induce are not yet completely understood in humans, especially when the effect is due to an environmental exposure dose. In general, the toxicity of POPs is influenced by many variables, including type, specific structure, dose, and way of exposure. One of the mechanisms of toxicity is mediated by aryl hydrocarbon receptor (AhR) activation, leading to the transcriptional activation of multiple genes including *CYP1A1* [47]. Other studies suggest that PBDEs may bind but not activate the AhR complex and subsequent transcriptional processes. As POPs act in mixtures with a large number of chemicals, the fact that both PBDEs and PCBs can be AhR antagonists could reduce the resulting effect of other dioxin-like compounds [48]. In addition, POPs may influence the effects of B[a]P and other PAHs, including their potential impacts on oxidative stress induction. If AhR antagonists predominate in the mixture, they inhibit AhR activity potentially resulting in lower-than-expected impacts of PAHs. This could explain the lack of association of B[a]P levels with oxidative stress markers observed in our study. Although the reduced metabolic activation of PAHs due to AhR inhibition could decrease ROS production, the processes associated with increased oxidative stress, production of pro-inflammatory cytokines, and excessive ROS generation could occur via the false mitochondrial oxidation of fats [49] caused by POPs [47].To reveal the mechanisms of biological impacts of BDE 99, BDE 154 and o,p′-DDE, found to be associated with oxidative stress markers in our study, in vitro mechanistic studies would be needed. In general, POPs are a group of chemical compounds that, due to their properties, must be used in strict accordance with safety standards and procedures, as many of them may induce oxidative stress or act as endocrine disruptors in humans.

## 5. Conclusions

In conclusion, we performed a thorough analysis of factors potentially affecting oxidative damage in the study populations. To the best of our knowledge, our report is the first that focused on associations between a comprehensive panel of POPs in blood plasma and oxidative stress markers in subjects originating from localities differing in environmental conditions. Despite this in-depth investigation, we were not able to identify a dominant biologically plausible mechanism that would contribute to the induction of oxidatively damaged DNA and lipid peroxidation in our study groups. Our data suggest that certain variables not identified here induced inflammation, resulting in pro-inflammatory cytokine production that most likely contributed to oxidative damage to macromolecules. Only weak antioxidant mechanisms were induced. The role of environmental contaminants monitored in this study, as well as the personal characteristics of the study subjects, was minor.

## Figures and Tables

**Figure 1 ijerph-19-03609-f001:**
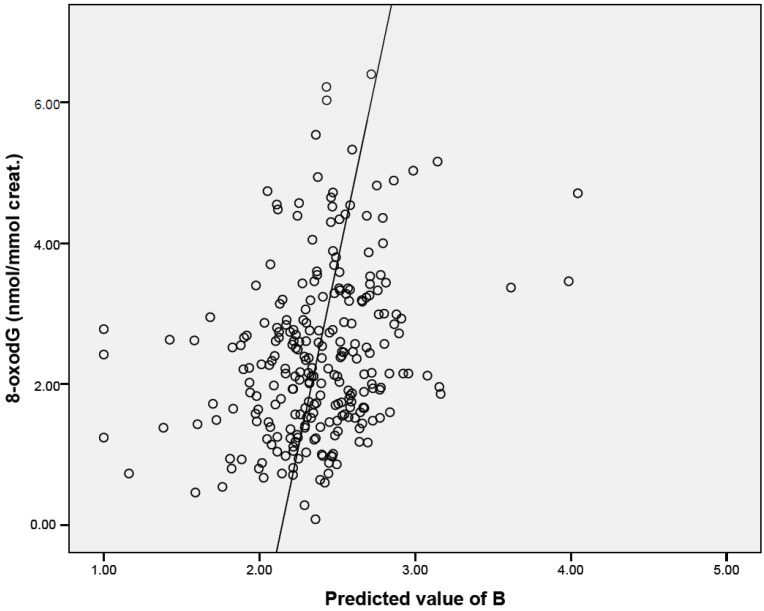
Graphical visualization of multivariate-adjusted association between 8oxodG levels and parameters reported in Table 5.

**Figure 2 ijerph-19-03609-f002:**
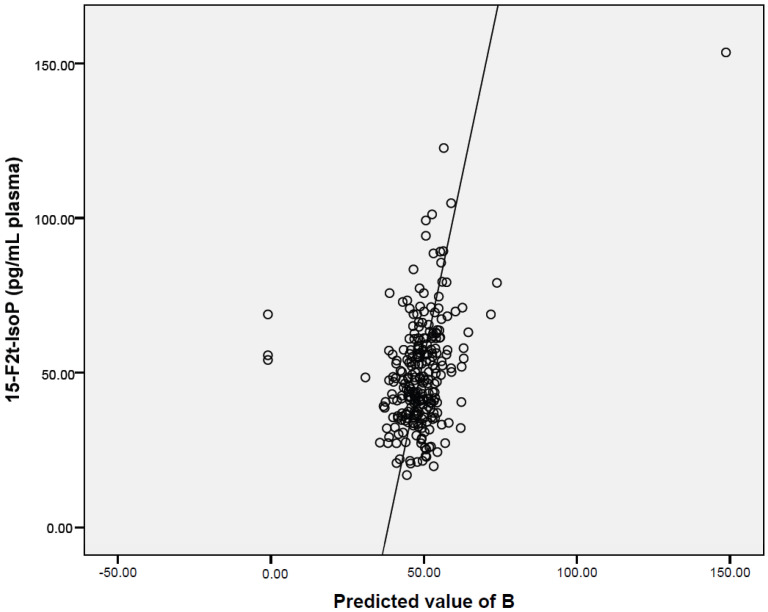
Graphical visualization of multivariate-adjusted association between 15-F2t-IsoP levels and parameters reported in Table 5.

**Table 1 ijerph-19-03609-t001:** Characteristics of the study population, exposure to environmental pollutants, parameters of antioxidant and immune response, and oxidative stress markers: locality Ceske Budejovice; SD–standard deviation.

Ceske Budejovice					
	Season 1 (N = 16)		Season 2 (N = 16)		
Variable	Mean ± SD	Median (Min, Max)	Mean ± SD	Median (Min, Max)	*p*-Value
Age (years)	38.0 ± 6.59	38.0 (22.0, 48.0)	38.4 ± 6.70	39.0 (22.0, 49.0)	0.87
BMI (kg/m^2^)	28.2 ± 3.88	28.1 (23.1, 41.0)	27.9 ± 4.00	27.3 (22.8, 41.0)	0.11
Cotinine (ng/mg creat.)	5.66 ± 3.51	4.93 (1.47, 13.8)	10.5 ± 29.1	3.33 (1.30, 120.0)	0.12
Education (high school/university) (N)	14/2		14/2		1.00
B[a]P (ng/m^3^)	0.23 ± 0.21	0.17 (0.05, 0.78)	0.33 ± 0.30	0.20 (0.06, 0.98)	0.31
PM_2.5_ (µg/m^3^)	7.48 ± 3.05	7.13 (4.85, 15.1)	4.73 ± 2.43	2.65 (2.65, 7.40)	0.14
Ozone (µg/m^3^)	56.3 ± 9.95	57.4 (37.4, 65.9)	38.6 ± 2.66	36.3 (36.3, 41.5)	<0.001
SOD (U/mL)	7.50 ± 1.80	8.16 (4.31, 10.1)	8.37 ± 3.06	10.0 (2.97, 12.2)	0.052
CAT (U/mL)	109.8 ± 17.7	109.3 (81.0, 143.0)	77.9 ± 10.8	76.6 (58.8, 96.9)	<0.001
GPx (U/mL)	139.2 ± 39.2	143.6 (25.1, 187.6)	83.1 ± 44.8	81.7 (16.2, 162.3)	<0.001
ORAC (µM TE)	5.03 ± 0.74	5.03 (4.10, 6.40)	4.68 ± 0.67	4.45 (3.90, 6.05)	0.16
TNF-α (pg/mL)	62.7 ± 91.9	6.20 (0, 250.0)	45.0 ± 83.8	1.88 (0, 250.0)	<0.01
IL-1β (pg/mL)	167.4 ± 73.0	152.7 (61.1, 303.1)	160.6 ± 152.9	108.9 (30.8, 500.0)	0.88
IL-6 (pg/mL)	7.96 ± 9.19	6.38 (0, 33.5)	0.96 ± 2.11	0 (0, 6.58)	<0.01
8-oxodG (nmol/mmol creat.)	2.11 ± 1.05	1.95 (0.73, 3.89)	1.57 ± 0.73	1.41 (0.28, 2.73)	0.04
15-F2t-IsoP (pg/mL plasma)	34.9 ± 9.14	34.6 (19.8, 47.6)	50.7 ± 10.2	48.6 (35.3, 75.7)	<0.001

**Table 2 ijerph-19-03609-t002:** Characteristics of the study population, exposure to environmental pollutants, parameters of antioxidant and immune response, and oxidative stress markers: locality Prague; SD–standard deviation.

Prague					
	Season 1 (N = 56)		Season 2 (N = 56)		
Variable	Mean ± SD	Median (Min, Max)	Mean ± SD	Median (Min, Max)	*p*-Value
Age (years)	39.5 ± 9.22	38.5 (23, 63)	39.9 ± 9.25	39.5 (23, 64)	0.80
BMI (kg/m^2^)	28.5 ± 3.89	29.2 (19.4, 36.8)	28.5 ± 3.80	29.0 (19.4, 36.8)	0.56
Cotinine (ng/mg creat.)	18.4 ± 69.9	5.55 (1.42, 502.4)	21.6 ± 63.6	5.04 (1.49, 390.8)	0.77
Education (high school/university) (N)	43/13		42/14		1.00
B[a]P (ng/m^3^)	0.23 ± 0.13	0.20 (0.05, 0.61)	0.59 ± 0.32	0.56 (0.13, 1.67)	<0.001
PM_2.5_ (µg/m^3^)	22.2 ± 9.19	27.1 (7.79, 32.0)	21.7 ± 9.44	21.7 (11.6, 35.9	0.62
Ozone (µg/m^3^)	68.2 ± 9.46	68.4 (6.20, 76.3)	44.3 ± 25.0	58.4 (6.20, 66.3)	<0.001
SOD (U/mL)	11.8 ± 7.34	8.91 (2.81, 31.1)	14.0 ± 37.7	6.11 (2.03, 207.9)	<0.01
CAT (U/mL)	88.2 ± 14.3	89.6 (50.4, 112.0)	86.4 ± 18.0	83.6 (58.8, 157.5)	0.15
GPx (U/mL)	157.6 ± 32.8	153.2 (95.0, 220.1)	124.7 ± 62.3	125.4 (5.36, 274.9)	<0.001
ORAC (µM TE)	4.65 ± 0.73	4.60 (3.28, 6.43)	4.80 ± 0.77	4.68 (3.28, 6.28)	0.29
TNF-α (pg/mL)	29.0 ± 60.7	6.34 (0, 250.0)	25.1 ± 61.0	2.84 (0, 250.0)	<0.01
IL-1β (pg/mL)	121.6 ± 97.5	90.3 (33.0, 500.0)	194.2 ± 126.3	151.4 (35.8, 500.0)	<0.001
IL-6 (pg/mL)	7.61 ± 10.2	4.32 (0, 65.7)	11.1 ± 15.0	5.26 (0, 68.3)	0.25
8-oxodG (nmol/mmol creat.)	2.56 ± 1.25	2.37 (0.08, 6.40)	2.79 ± 1.20	2.53 (0.98, 6.22)	0.20
15-F2t-IsoP (pg/mL plasma)	50.4 ± 15.0	48.9 (16.9, 88.6)	45.7 ± 13.8	43.85 (20.6, 70.8)	0.01

**Table 3 ijerph-19-03609-t003:** Characteristics of the study population, exposure to environmental pollutants, parameters of antioxidant and immune response, and oxidative stress markers: locality Ostrava; SD–standard deviation.

Ostrava					
	Season 1 (N = 54)		Season 2 (N = 54)		
Variable	Mean ± SD	Median (Min, Max)	Mean ± SD	Median (Min, Max)	*p*-Value
Age (years)	40.4 ± 9.37	42.0 (21.0, 61.0)	40.9 ± 9.35	43.0 (22.0, 62.0)	0.81
BMI (kg/m^2^)	28.6 ± 4.12	28.4 (20.4, 44.8)	28.5 ± 4.21	28.4 (21.1, 46.3)	0.49
Cotinine (ng/mg creat.)	8.28 ± 8.57	5.69 (1.66, 47.3)	46.3 ± 243.4	4.56 (0.58, 1789)	0.96
Education (high school/university) (N)	50/4		50/4		1.00
B[a]P (ng/m^3^)	0.43 ± 0.75	0.29 (0.08, 5.18)	0.17 ± 0.15	0.11 (0.05, 0.68)	<0.001
PM_2.5_ (µg/m^3^)	5.91 ± 1.88	5.70 (3.53, 9.33)	6.84 ± 2.11	6.50 (4.58, 11.2)	0.03
Ozone (µg/m^3^)	50.1 ± 16.0	51.5 (5.60, 69.5)	15.2 ± 11.1	10.8 (4.60, 31.4)	<0.001
SOD (U/mL)	14.9 ± 8.9	14.1 (4.65, 69.7)	8.39 ± 5.35	7.25 (2.54, 39.3)	<0.001
CAT (U/mL)	96.5 ± 14.1	97.6 (59.0, 124.1)	97.6 ± 17.1	97.4 (54.9, 139.1)	0.72
GPx (U/mL)	113.4 ± 35.2	11.0 (18.5, 195.0)	81.8 ± 62.4	59.7 (5.03, 222.3)	<0.01
ORAC (µM TE)	5.12 ± 1.07	4.97 (3.13, 7.71)	4.58 ± 0.63	4.57 (3.21, 5.97)	0.002
TNF-α (pg/mL)	39.2 ± 73.6	2.98 (0, 250.0)	33.5 ± 66.5	1.75 (0, 250.0)	0.25
IL-1β (pg/mL)	229.7 ± 110.2	204.1 (98.3, 500.0)	173.5 ± 145.6	120.4 (10.4, 500.0)	<0.01
IL-6 (pg/mL)	7.40 ± 5.45	6.13 (0, 19.5)	62.7 ± 145.5	20.2 (0, 908.8)	<0.001
8-oxodG (nmol/mmol creat.)	2.25 ± 1.16	2.27 (0.46, 6.03)	2.24 ± 1.04	2.11 (0.60, 4.89)	0.96
15-F2t-IsoP (pg/mL plasma)	53.4 ± 26.1	45.3 (21.5, 153.6)	52.7 ± 15.7	51.7 (25.2, 104.8)	0.49

**Table 4 ijerph-19-03609-t004:** A comparison of population characteristics, environmental pollutants, antioxidant and inflammatory response, and oxidative stress markers for individual seasons across the localities.

Variable	Season 1 (*p*-Value)	Season 2 (*p*-Value)
	Personal characteristics	
Age	0.61	0.60
BMI	0.70	0.55
Cotinine	0.45	0.04
Education	0.07	0.04
Antioxidant response parameters		
SOD	<0.001	0.12
CAT	<0.001	<0.001
GPx	<0.001	<0.001
ORAC	0.02	0.26
Proinflammatory response		
TNF-α	0.39	0.74
IL-1β	<0.001	0.09
IL-6	0.48	<0.001
Air pollutants		
B[a]P	<0.01	<0.001
PM_2.5_	<0.001	<0.001
Ozone	<0.001	<0.001
Polychlorinated biphenyls		
PCB 28	0.09	0.26
PCB 52	<0.001	0.08
PCB 101	0.21	0.02
PCB 118	0.08	0.49
PCB 138	0.04	0.13
PCB 153	0.01	0.03
PCB 170	<0.001	0.04
PCB 180	<0.001	0.01
Organochlorinated pesticides		
o,p′-DDE	0.01	0.22
p,p′-DDE	0.15	0.19
o,p′-DDD	0.12	0.07
p,p′-DDD	0.01	0.04
o,p′-DDT	0.02	0.65
p,p′-DDT	0.02	0.10
HCB	0.23	0.74
α-HCH	<0.001	0.26
β-HCH	0.73	0.01
γ-HCH	0.02	0.18
Brominated flame retardants		
BDE 47	<0.001	0.01
BDE 99	0.52	0.11
BDE 100	0.65	0.91
BDE 153	0.24	0.08
BDE 154	1.00	0.86
BDE 183	0.63	0.48
BDE 209	<0.01	0.12
Per- and polyfluoroalkylated substances		
PFBS	0.54	0.25
PFH×S	<0.01	<0.01
PFOS	0.46	0.06
PFDS	1.00	0.22
PFBA	0.69	0.08
PFHpA	0.04	0.10
PFOA	0.83	0.31
PFNA	0.05	0.04
PFDA	<0.01	<0.001
PFUdA	0.29	<0.01
PFDoA	0.09	0.06
PFTrDA	0.21	0.13
PFTeDA	0.17	0.03
Monohydroxylated PAH metabolites		
1-OH-NAP	0.45	<0.01
2-OH-NAP	0.76	0.55
2-OH-FLUO	0.33	<0.01
1-OH-PHEN	<0.001	<0.01
2-OH-PHEN	0.02	<0.01
3-OH-PHEN	<0.01	<0.001
4-OH-PHEN	<0.01	<0.01
9-OH-PHEN	0.79	0.11
1-OH–pyrene	<0.001	<0.001
Oxidative stress markers		
8-oxodG	0.26	<0.001
15-F2t-IsoP	<0.01	0.04

**Table 5 ijerph-19-03609-t005:** Multivariate estimates of associations between oxidative stress markers and selected parameters.

	8-oxodG		15-F2t-IsoP		
	B *, 95% CI	*p*-Value		B *, 95% CI	*p*-Value
Age	0.01 (−0.004, 0.03)	0.12	Age	0.01 (−0.32, 0.34)	0.95
BMI	−0.02 (−0.06, 0.01)	0.21	BMI	−0.42 (−0.98, 0.13)	0.13
Cotinine	0.001 (0.00, 0.002)	0.03	Cotinine	−0.02 (−0.02, −0.01)	<0.001
Education	−0.33 (−0.81, 0.15)	0.18	Education	−0.07 (−6.22, 6.09)	0.98
Locality	0.04 (−0.20, 0.27)	0.78	Locality	2.81 (−0.24, 5.85)	0.07
SOD	0.00 (−0.01, 0.009)	0.98	SOD	0.05 (−0.09, 0.18)	0.50
CAT	−0.002 (−0.01, 0.008)	0.66	CAT	0.18 (0.06, 0.31)	0.005
GP×	0.003 (0.00, 0.005)	0.05	GPX	−0.009 (−0.04, 0.02)	0.58
ORAC	−0.23 (−0.43, −0.04)	0.02	ORAC	−0.65 (−3.37, 2.06)	0.64
TNF-α	0.001 (−0.001, 0.003)	0.40	TNF-α	−0.01 (−0.05, 0.02)	0.43
IL-1β	0.00 (−0.002, 0.001)	0.73	IL-1β	0.01 (−0.005, 0.03)	0.15
IL-6	0.002 (0.001, 0.004)	<0.001	IL-6	0.02 (0.008, 0.04)	0.003
o,p′-DDE	−0.005 (−0.007, −0.004)	<0.001	BDE 154	−28.4 (−33.4, −23.5)	<0.001
BDE 154	−0.97 (−1.75, −0.19)	0.02	BDE 99	3.93 (3.44, 4.43)	<0.001

* Adjusted to variables reported in the table; 95% CI—95% confidence interval.

## Data Availability

Data are contained within the article or Appendix A.

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
