# Peer review of "Oxidative Stress and Antioxidant Response in Populations of the Czech Republic Exposed to Various Levels of Environmental Pollutants"

_ijerph, 2022, doi:10.3390/ijerph19063609_

Round 1

Reviewer 1 Report

In this work, authors focused on the oxidative damage of DNA and lipid peroxidation caused by different air pollution. The authors conducted a considerable amount of data collection to try to reveal the connection by oxidative stress parameters, which seems interesting and significant. However, the manuscript could not be recommended for publication before addressing the following major concerns.

  1. By contrast with the Introduction part, some information in the Discussion part tended to repetition. It is suggested to reorganize the information and the framework in the manuscript.
  2. It is suggested to perform some figures like multivariable linear regression analysis to display the data more clearly and directly.
  3. Authors should state more clearly about the other mechanisms or reactions that mediate antioxidant defense besides the enzymes.
  4. Please expand the “PDNA” mentioned in page 8, line 327.

     5. The authors can discuss more advantages and innovation about the                work, instead of just stating the failure in the Conclusion part. 

Author Response

Response to Reviewers’ comments

We thank the Reviewers for their valuable comments that helped to improve our manuscript. We have carefully checked the points raised by the Reviewers and modified the text as requested. Please, find the response to the comments below.

Reviewer 1

In this work, authors focused on the oxidative damage of DNA and lipid peroxidation caused by different air pollution. The authors conducted a considerable amount of data collection to try to reveal the connection by oxidative stress parameters, which seems interesting and significant. However, the manuscript could not be recommended for publication before addressing the following major concerns.

  1. By contrast with the Introduction part, some information in the Discussion part tended to repetition. It is suggested to reorganize the information and the framework in the manuscript.

Response: We have carefully checked both parts of the manuscript and removed repeated information in Discussion.

  1. It is suggested to perform some figures like multivariable linear regression analysis to display the data more clearly and directly.

Response: As there are many parameters in the multivariate linear regression that impact oxidative stress markers, simple graphical presentation is rather difficult. To show the trends of multivariate-adjusted associations between independent variables and 8-oxodG/15-F2t-isoprostane levels, we prepared scatter plots for each oxidative stress marker in which levels of these markers are plotted against predicted values of B parameters of multivariate linear regression and we report them as Figure 1 and 2. We also added description of this approach to Materials and Methods/Statistical analysis section.

  1. Authors should state more clearly about the other mechanisms or reactions that mediate antioxidant defense besides the enzymes.

Response: We added information on nonenzymatic and repair systems to Discussion to complement information on antioxidant enzymes presently reported in the manuscript.

  1. Please expand the “PDNA” mentioned in page 8, line 327.

Response: This is a typo, thank you for pointing it out. The abbreviation should read PFDA (perfluoro-n-decanoic acid)

  1. The authors can discuss more advantages and innovation about the work, instead of just stating the failure in the Conclusion part.

Response: As suggested, we added more details about advantages of our study to the Conclusions.

Reviewer 2 Report

As a reviewer I have the following remarks. I am using “you” as Dear Authors.

  1. Line 59: “COPD” – full spell in the first use. If only one time used the abbreviation, don’t introduce it. Also “PM2.5” – 2.5 should be subscript, it’s OK, but in a final/galley proof version should be corrected.
  2. Line 260. You can add in (IBM SPSS Statistics for Windows, Version 20.0).
  3. My remarks on the tables (1-3). We see the abbreviation SD – should be added an explanation. Also the precision the number of numerals after dot – In my opinion on is OK (for example for Age), As we see different length it confuses a potential reader. Probably to keep the same ” 40.4 ± 9.37’, I think it’s better: 40.4 ± 9.4 (rounded). But this “ 0.43 ± 0.75” makes sense.
  4. Under Table % please add: 95% CI – 95% confidence interval.
  5. Line 383: “policemen spending most of their shifts outdoors.” – in the study do we know the patter of their indoor/outdoor activities related to study period?
  6. Line 524: “Data is contained” – “are”, data are plural.
  7. Any reason that ozone was not considered?

Thank you

Author Response

Response to Reviewers’ comments

We thank the Reviewers for their valuable comments that helped to improve our manuscript. We have carefully checked the points raised by the Reviewers and modified the text as requested. Please, find the response to the comments below.

Reviewer 2

As a reviewer I have the following remarks. I am using “you” as Dear Authors.

  1. Line 59: “COPD” – full spell in the first use. If only one time used the abbreviation, don’t introduce it. Also “PM2.5” – 2.5 should be subscript, it’s OK, but in a final/galley proof version should be corrected.

Response: The changes were made as suggested.

  1. Line 260. You can add in (IBM SPSS Statistics for Windows, Version 20.0).

Response: Corrected as requested.

  1. My remarks on the tables (1-3). We see the abbreviation SD – should be added an explanation. Also the precision the number of numerals after dot – In my opinion on is OK (for example for Age), As we see different length it confuses a potential reader. Probably to keep the same ” 40.4 ± 9.37’, I think it’s better: 40.4 ± 9.4 (rounded). But this “ 0.43 ± 0.75” makes sense.

Response: We added the explanation of “SD” (standard deviation). Regarding the number of decimal points, we tried to keep the same number of significant figures, i.e. the number of digits in each value. For most values, this number is 3 (i.e. 28.2 or 3.88), except values greater than 100, where 4 significant figures were used. To present the data in a consistent way, we suggest to keep this logic of decimal points presentation. To follow this way of presentation, we did some minor modifications in the tables.

  1. Under Table % please add: 95% CI – 95% confidence interval.

Response: The text was added.

  1. Line 383: “policemen spending most of their shifts outdoors.” – in the study do we know the patter of their indoor/outdoor activities related to study period?

Response: For some subjects, we have the data on time spent outdoors and indoors. However, as this information is not consistent and not available for all study participants, we were not able to include it in analyses of impacts of environmental factors on oxidative stress markers.

  1. Line 524: “Data is contained” – “are”, data are plural.

Response: The text was corrected.

  1. Any reason that ozone was not considered?

Response: We thank the Reviewer for this important comment. We included information on ozone concentrations in individual study seasons (Table 1-3) and analyzed associations between oxidative stress markers and ozone levels. However, as we did not see any significant result, this variable was not included in multivariate analyses. Thus, in our study, the impact of ozone concentrations on oxidative stress seems to be limited. The information on analyses of ozone impacts was added to Results/Discussion.

Round 2

Reviewer 1 Report

This work can be publicated after revision.